# Overcoming Vocabulary Mismatch:
# Vocabulary-agnostic Teacher Guided Language Modeling

**Haebin Shin** [1 2 *]  **Lei Ji** [1]  **Xiao Liu** [1]  **Yeyun Gong** [1]

## Abstract

Using large teacher models to guide the training of smaller student models has become the prevailing paradigm for efficient and effective learning. However, vocabulary mismatches between teacher and student language models pose significant challenges in language modeling, resulting in divergent token sequences and output distributions. To overcome these limitations, we propose Vocabulary-agnostic Teacher Guided Language Modeling (VocAgnoLM), a novel approach that bridges the gap caused by vocabulary mismatch through two key methods: (1) Token-level Lexical Alignment, which aligns token sequences across mismatched vocabularies, and (2) Teacher Guided Loss, which leverages the loss of teacher model to guide effective student training. We demonstrate its effectiveness in language modeling with 1B student model using various 7B teacher models with different vocabularies. Notably, with Qwen2.5-Math-Instruct, a teacher model sharing only about 6% of its vocabulary with TinyLlama, VocAgnoLM achieves a 46% performance improvement compared to naive continual pretraining. Furthermore, we demonstrate that VocAgnoLM consistently benefits from stronger teacher models, providing a robust solution to vocabulary mismatches in language modeling.

## 1. Introduction

Large language models (LLMs) have increasingly adopted *guidance from teacher* models to enhance student LLM

---
[*]Work done during internship at Microsoft Research.
[1]Microsoft Research [2]KAIST AI. Correspondence to: Haebin Shin <haebin.shin@kaist.ac.kr>, Lei Ji <leiji@microsoft.com>, Xiao Liu <xiao.liu.msrasia@microsoft.com>, Yeyun Gong <yegong@microsoft.com>.

*Proceedings of the $42^{nd}$ International Conference on Machine Learning*, Vancouver, Canada. PMLR 267, 2025. Copyright 2025 by the author(s).

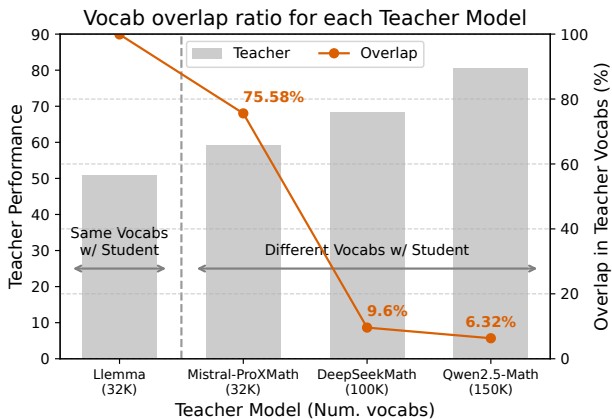

Figure 1: Limitation in Utilizing Better LLMs as Teacher Models due to Vocabulary Mismatch: Qwen2.5-Math (Yang et al., 2024) outperforms Llemma (Azerbayev et al., 2024) on math evaluation suite, but shares only 6.32% of its vocabulary with the student model, TinyLlama (Zhang et al., 2024a).

training. This paradigm has been instrumental in addressing critical challenges, such as compensating for the limited capacity of smaller models during pretraining (Gemma-Team et al., 2024; Meta, 2024; Muralidharan et al., 2024), optimizing language modeling with carefully selected or curated data (Gu et al., 2024b; Lin et al., 2024), and providing task-specific on-policy guidance for downstream tasks (Gu et al., 2024a; Agarwal et al., 2024). By inheriting capabilities from more advanced models, student LLMs can refine their behaviors, align with specific objectives, and achieve substantial performance gains.

Despite these advances, a critical bottleneck remains: the **vocabulary mismatch** between teacher and student models. Most current methods assume identical vocabularies, limiting the student model's ability to benefit from the latest state-of-the-art or domain-specialized teacher models with distinct tokenization schemes. This constraint not only prevents student models from leveraging diverse open-source LLMs but also imposes additional costs, such as the need to train customized teacher models or rely exclusively on large models with compatible vocabularies (Gu et al., 2024b; Lin et al., 2024). As illustrated in Figure 1, the vocabu-

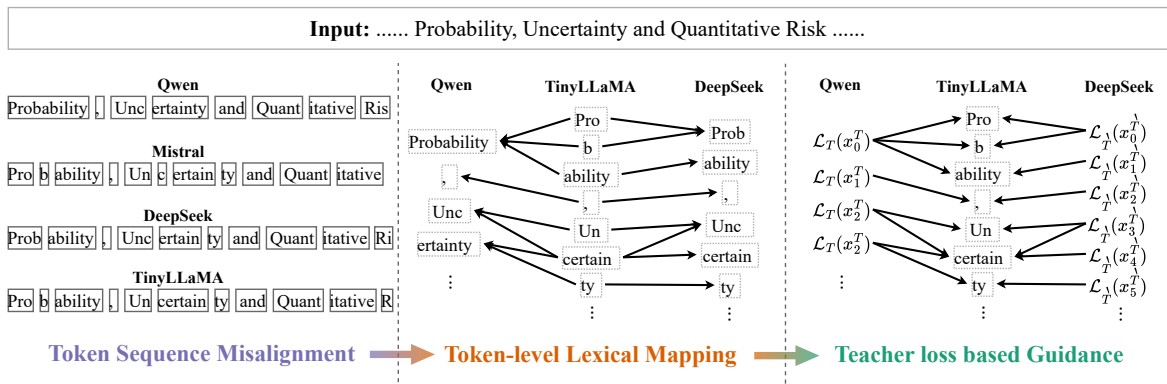

Figure 2: Overview of Vocabulary-agnostic Teacher Guided Language Modeling. *Left*: Teacher models (such as Qwen, Mistral, DeepSeek) produce token sequences that differ from those of the student model (TinyLlama), leading to misalignment. *Middle*: To address this, Token-level Lexical Mapping establishes a one-to-many mapping from each student token to corresponding teacher tokens. *Right*: To overcome logit distribution divergence, the mapped teacher token loss is utilized to guide the training of the student model.

lary overlap ratio between Qwen (Qwen et al., 2024) and TinyLlama (Zhang et al., 2024a) is only 6.32%, despite Qwen's strong performance. This disparity underscores how vocabulary mismatches severely limit the adoption of high-performing teacher models, creating an urgent need for approaches that transcend these restrictions.

Given this mismatch, we analyze the tokenization discrepancy between different LLMs to observe two primary limitations: **(1) token sequence mismatch** and **(2) logit distribution divergence**. As illustrated in Figure 2, different LLMs tokenize the same input phrase `"Probability, Uncertainty"` into 4, 8, 6, and 7 tokens, respectively. This tokenization variability disrupts sequence alignment, complicating the student model's ability to interpret the teacher model's outputs. Furthermore, even if two models produce a similar number of tokens, their logit distributions can vary significantly due to differences in architecture, training data, or optimization techniques. This makes it challenging to directly use teacher model's logits as guidance for student model.

To overcome these challenges, we propose a simple yet effective approach, **Token-level Lexical Alignment**. Our method achieves token-level alignment in a one-to-many manner, enabling student models to receive fine-grained guidance from teacher models without requiring additional training to unify vocabularies. By bridging the gap between differing vocabularies, Token-level Lexical Alignment allows the efficient integration of emerging teacher models and ensures the student model can effectively learn from teacher models' guidance. Additionally, we introduce **Teacher Guided Loss** to leverage aligned tokens, allowing the student to benefit from the aligned teacher outputs, even when their logit distributions differ.

We evaluate VocAgnoLM by continual pretraining TinyL-LaMA 1.1B (Zhang et al., 2024a) using 7B teacher models built on a different vocabulary system, such as Mistral (Jiang et al., 2023), DeepSeek (DeepSeek-AI et al., 2024), or Qwen2.5 (Qwen et al., 2024). Notably, using Qwen2.5-Math-Instruct (Yang et al., 2024), which has only 6% vocabulary overlap with TinyLLaMA, VocAgnoLM achieves a 46% performance improvement over naive continual pretraining baseline and a 33% improvement over a a logit distribution mapping baseline.

We highlight the effectiveness of vocabulary-agnostic distillation, enabling specialized teacher models to be directly utilized without vocabulary constraints. Our approach not only mitigates the vocabulary mismatch issue but also enhances the efficiency and performance of domain-specific language modeling tasks.

Our contributions are summarized in three folds:

- We tackle the vocabulary mismatch problem in the language modeling domain, which has remained underexplored despite its critical impact on leveraging teacher guidance. To address this, we introduce **Vocabulary-agnostic Teacher Guided Language Modeling (VocAgnoLM)**, a novel approach that bridges vocabulary discrepancies, enabling effective cross-vocabulary teacher guidance.

- We propose a **Token-level Lexical Alignment** approach that enables fine-grained, token-by-token guidance to address sequence mismatches between student and teacher tokens. Furthermore, we employ a **teacher loss based guidance** to address the logit distribution gap, ensuring the student model's effective training across vocabularies mismatch.

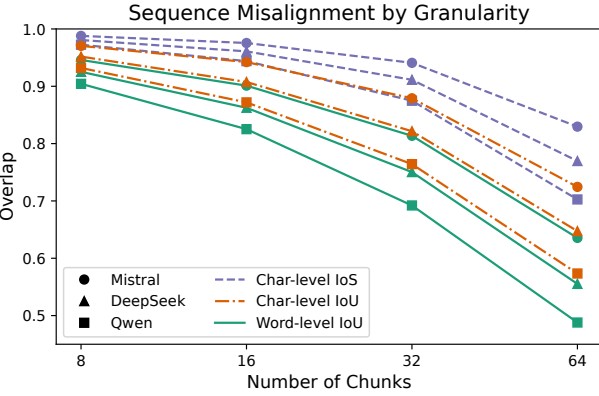

Figure 3: Comparison of Sequence Overlap by Granularity. Sequence overlap between the corresponding chunks of student (TinyLlama) and teacher models differs significantly across varying levels of granularity (Number of Chunks). IoU (Intersection over Union) refers to the overlap ratio between the two sequences, while IoS (Intersection over Student sequence) denotes the coverage of the student sequence by the teacher sequence.

- We demonstrate that our method yields performance improvements proportional to the strength of the teacher model, regardless of vocabulary mismatch. This flexibility allows new, high-performing teacher models to be seamlessly integrated without the need for vocabulary compatibility, showcasing the practicality of our approach.

## 2. Preliminary Study

Our motivation begins with the challenge of aligning two token sequences that have been segmented differently by a teacher and a student model. According to the showcase in Figure 2, various tokenizers generate quite different length of token sequence. A straightforward approach to align them is to divide both sequences into an equal number of coarse-grained chunks (Xie et al., 2024; Xu et al., 2025). We first equally chunk each sequence into a predefined number of chunks and then evaluate the Intersection over Union (IoU) and Intersection over Student sequence (IoS) for each pair of mapped chunks to assess the quality of the mapping.

As illustrated in Figure 3, we observe the degree of alignment mismatch at various levels of granularity to understand how these two differently segmented sequences diverge. As we increase the number of chunks (i.e., move toward a finer-grained segmentation), the alignment between the two sequences deteriorates. On the one hand, small number of chunks shows higher overlap, however, this leads to a coarse-grained mapping. On the other hand, fine-grained mapping with large number of chunks exhibits lower overlap, due to the equally chunking strategy. As shown in Figure 3, this

leads to a progressive decrease in the coverage of student tokens by teacher chunks, thereby making it increasingly difficult for the student to receive precise guidance from the teacher. This motivates us to design comprehensive alignment algorithm, where the student can receive token-level guidance from fine-grained teacher token alignment, ensuring precise and effective supervision.

## 3. Vocabulary-agnostic Teacher Guided Language Modeling

We introduce Vocabulary-agnostic Teacher Guided Language Modeling (VocAgnoLM) to address vocabulary mismatch between a student model ($S$) and a teacher model ($T$). To tackle the *token sequence mismatch*, and *logit distribution divergence*, we propose Token-level Lexical Alignment (Section 3.1) for sequence alignment and Teacher Guided Loss (Section 3.2) to enable effective teacher guidance despite differing logit distributions.

### 3.1. Token-level Lexical Alignment

Our primary objective is to enable the student model to receive fine-grained, token-level guidance from a teacher model during language modeling. The challenge arises when the student and teacher models have different vocabularies, leading to discrepancies in how the same text span is split into tokens. For example, a single student token may span multiple teacher tokens or vice versa due to their unique vocabulary sets. To address this, we leverage character-level offsets $[st, ed]$ representing the starting and end positions of a token, which can precisely map the tokens between two sequences according to the original position in the raw text.

Let us denote student tokens $\{x_1^S, x_2^S, \ldots, x_N^S\}$ with character offsets $[st_i^S, ed_i^S]$ for each $x_i^S$, and teacher tokens $\{x_1^T, x_2^T, \ldots, x_M^T\}$ with character offsets $[st_j^T, ed_j^T]$ for each $x_j^T$. By tracking these offsets, we can determine the exact text span covered by a student token $x_i^S$ and identify the corresponding teacher tokens $\{x_j^T, \ldots, x_k^T\}$ that fully cover that span. Specifically, for each student token $x_i^S$, we define the mapping function ($\mathrm{mapping}[i]$) to be the index range of teacher tokens that covers $x_i^S$ as follow as:

$$\mathrm{mapping}[i] = \begin{cases} -1, & \text{if none teacher token covers } x_i^S, \\ (j, k), & \text{if } \quad x_i^S \subseteq x_{[j,k]}^T \\ & \qquad \wedge \ x_i^S \not\subseteq x_{[j+1,k]}^T \\ & \qquad \wedge \ x_i^S \not\subseteq x_{[j,k-1]}^T \end{cases}$$

Here, $x_{[j,k]}^T$ denotes the concatenation of teacher tokens $x_j^T, x_{j+1}^T, \ldots, x_k^T$. The conditions $x_i^S \not\subseteq x_{[j+1,k]}^T \ \wedge \ x_i^S \not\subseteq x_{[j,k-1]}^T$ ensure that $(j, k)$ is the minimal index range that covers $x_i^S$. And, $\mathrm{mapping}[i] = -1$ corresponds to an un-

**Algorithm 1** Token-level Lexical Alignment
  **Input:**
     Student tokens $\{x_i^S\}$ with offsets $\{[st_i^S, ed_i^S]\}$
     Teacher tokens $\{x_j^T\}$ with offsets $\{[st_j^T, ed_j^T]\}$
  **Output:** Mapping mapping[$i$]
  **for** each $i$-th student token $x_i^S$ **do**
    (1) Get student character-level offsets:
       $st_i^S, ed_i^S \leftarrow x_i^S$
    (2) Find overlapping teacher range via binary searches:
       $lowIdx \leftarrow$ lower bound of $\{\, t \mid ed_t^T > st_i^S \,\}$
       $highIdx \leftarrow$ upper bound of $\{\, t \mid st_t^T < ed_i^S \,\} - 1$
       $(j, k) \leftarrow (lowIdx, highIdx)$
    (3) Map corresponding teacher tokens: $x_{[j,k]}^T$
       mapping[$i$] $\leftarrow \{j, j+1, \ldots, k\}$
  **end for**
  **return** mapping

mapped student token, which is not lexically covered by any teacher tokens.

These mappings can be efficiently determined by performing two binary searches for each student token: one to find the earliest teacher token index (*lowIdx*) whose end offset exceeds $st_i^S$, and another to find the latest teacher token index (*highIdx*) whose start offset remains below $ed_i^S$. The detailed procedure is outlined in Algorithm 1. Since each of these range searches can be completed in $O(\log M)$ time, where M is the number of teacher tokens, the overall complexity for all N student tokens is $O(N \log M)$.

By defining a clear token alignment procedure, we establish a one-to-many mapping (one student token $x_i^S$ potentially corresponding to multiple teacher tokens $x_{[j,k]}^T$). This alignment enables the student model to more precisely leverage the teacher model's outputs, facilitating a form of token-level supervision.

### 3.2. Teacher Guided Language Modeling

Although Token-level Lexical Alignment aligns the two token sequences, the mismatch in logit distribution remains a challenge due to the differing vocabularies of the student and teacher models. To address this challenge, we leverage the loss values of the mapped teacher tokens to guide the importance of student tokens (Fan & Jaggi, 2023; Lin et al., 2024).

Based on the Token-level Lexical Alignment described in Section 2.1, the causal language modeling for the $i$-th student token ($x_i^S$) and the corresponding token losses for the teacher tokens ($x_{[j,k]}^T$) spanning from $j$ to $k$ are defined as follows:

$$\mathcal{L}_S(x_i^S) = -\log P(x_i^S \mid x_{<i}^S) \qquad (1)$$

$$\mathcal{L}_T(x_{[j,k]}^T) = \Phi_{l \in [j,k]}\Big[-\log P(x_l^T \mid x_{<l}^T)\Big] \qquad (2)$$

Here, $\Phi$ represents an aggregation function (e.g. summation, maximum, or mean), applied to the teacher model's token loss over the range $[j, k]$.

We utilize the mapped teacher token losses to guide the reweighting of the student token's importance, as defined in 3.

$$\mathcal{L}(S) = -\mathbb{E}_{i \sim [1,N]}\Big[\mathcal{W}(x_i^S) \cdot \log P(x_i^S \mid x_{<i}^S; S)\Big] \quad (3)$$

Specifically, $\mathcal{W}(x_i^S)$ determines whether a given token is adopted based on the difference between the student and the corresponding teacher token losses, which reflects the importance of the token (Fan & Jaggi, 2023; Lin et al., 2024). We apply a top-k threshold to identify the most important tokens, ensuring that only tokens with significant contributions are retained, along with unmapped student tokens (further discussed in Section 6.2).

$$\mathcal{W}(x_i^S) = \begin{cases} 1, & \text{if } L_S(x_i^S) - L_T(x_{[j,k]}^T) \in \text{Threshold} \\ & \quad \text{or} \quad \text{mapping}[i] == -1, \\ 0, & \text{otherwise} \end{cases}$$

$$(4)$$

Teacher token loss-based guidance enables the student model to receive guidance from the teacher model even when their vocabulary spaces differ, as each model computes its logit dimensions independently.

## 4. Experimental Setup

### 4.1. Dataset

**Pretraining Corpus.** We utilize OpenWebMath (Paster et al., 2024), containing about 15 billion tokens sourced from math-related web pages in the Common Crawl.

**Evaluation Setup.** To evaluate performance, we assess the model on 9 mathematical reasoning benchmarks covering diverse domains, question formats (multiple-choice and open-ended), and difficulty levels (elementary to university): GSM8k (Cobbe et al., 2021), MATH (Hendrycks et al., 2021b), GSM-Hard (Gao et al., 2022), SVAMP (Patel et al., 2021), ASDiv (Miao et al., 2020), MAWPS (Koncel-Kedziorski et al., 2016), TabMWP (TAB) (Lu et al., 2023), MathQA (MQA) (Amini et al., 2019), MMLU-STEM (Hendrycks et al., 2021a), and SAT (Azerbayev et al., 2024). We utilize few-shot chain-of-thought (CoT) examples (Wei et al., 2022) following the settings in Lin et al. (2024); Zhou et al. (2024).

### 4.2. Baselines

**KLD.** When the teacher model shares the same vocabulary, one of the most representative teacher guided language modeling is knowledge distillation based on Kullback–Leibler Divergence (KLD). Following Song et al.

(2020); Gu et al. (2024a), we incorporates the teacher-student logit KL divergence term to student model's cross entropy as defined in Equation (1). Detailed equation is described in Appendix B.

**ULD.** Boizard et al. (2025) introduce the Universal Logit Distillation (ULD) loss, designed to align the probability distributions of a student model and a teacher model with differing vocabularies. ULD loss minimizes the Wasserstein distance between the two distributions during finetuning on various downstream tasks, serving as an alternative to KL divergence. In this work, we compare the ULD loss-based logit distribution alignment with our teacher token loss-based guidance in the context of language modeling. Detailed equation is described in Appendix B.

**Rho-1.** Lin et al. (2024) utilize guidance from a well-curated oracle reference model with a shared vocabulary (e.g., trained on GPT-generated datasets or targeted corpora) to enable efficient language modeling through various scoring methods. In this study, we compare their approach, which employs the TinyLlama architecture as the reference model initially trained on OpenWebMath (Paster et al., 2024), using multi-criteria scoring based on token entropy and loss delta.

### 4.3. Implementation Details

We use LitGPT (Lightning-AI, 2023) to continually pretrain on 15B tokens from OpenWebMath (Paster et al., 2024). Training is conducted on 32 H100 GPUs with a cosine learning rate scheduler (decaying from 8e-5 to 8e-6), a sequence length of 2048, and a global batch size of 2M tokens, following prior works (Zhang et al., 2024a; Lin et al., 2024; Zhou et al., 2024). We apply a top-k threshold of 40%. Details are described in Appendix B.

**Models.** We conduct continual pretraining using TinyL-lama 1.1B (Zhang et al., 2024a), which has a vocabulary size of 32,000 tokens. To provide teacher guidance, we utilize 7B-scale, math-specialized teacher models: Llemma (Azerbayev et al., 2024), Mistral-ProXMath (Zhou et al., 2024), DeepSeekMath (Shao et al., 2024), and Qwen2.5-Math (Yang et al., 2024). Llemma (Azerbayev et al., 2024) shares the same vocabulary as TinyL-lama (Zhang et al., 2024a), and we use it as a teacher model to evaluate the impact of using a same vocabulary. Details of teacher models are provided in Appendix A. Table 4 reports the performance of the teacher models.

## 5. Experiments

### 5.1. Main Results

**Comparision with KLD (Same Vocabulary).** As shown in Table 1, compared to the KLD approach that fully utilizes

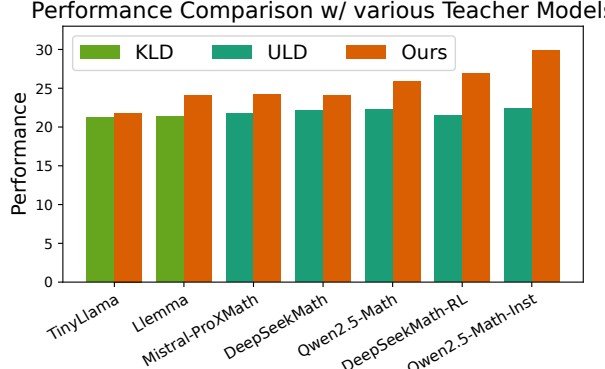

Figure 4: Performance Comparison Across Various Teacher Models. VocAgnoLM consistently outperforms logit distribution-based baselines.

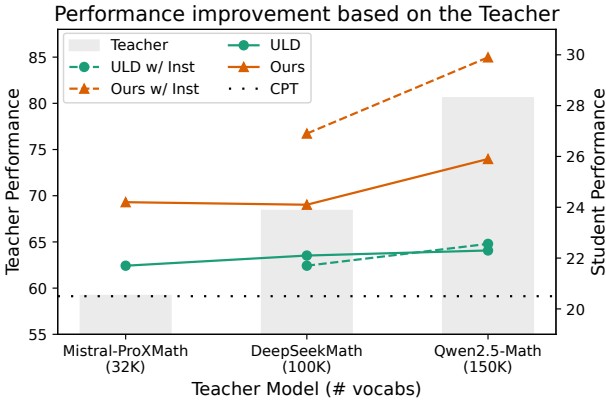

Figure 5: Comparison of Performance Improvements Across Different Teachers. VocAgnoLM effectively mitigates vocabulary mismatch and leverages higher-performing teacher models to achieve significant performance gains, outperforming logit distribution-based baselines.

the logit distribution, VocAgnoLM demonstrates superior performance. This difference becomes more pronounced when using a stronger teacher model, Llemma (Azerbayev et al., 2024), compared to a weaker teacher model (TinyLlama-CPT). By focusing solely on teacher guidance under the same aligned token sequence, our teacher loss-based guidance effectively facilitates student model training.

**Comparison with ULD (Different Vocabulary).** The limitations of probabilistic distribution-based distillation become more pronounced when using teacher models with different vocabularies. As shown in Figure 4, VocAgnoLM significantly outperforms ULD (Boizard et al., 2025), with Table 1 highlighting a substantial 33% performance gap when using Qwen2.5-Math-Instruct (Yang et al., 2024) as the teacher model. In the pretraining stage, where a large vol-

Table 1: Performance Comparison of Student Model ($S$) Guided by Various Teacher Models. [†]Average scores for comparison with the Rho-1, following Lin et al. (2024) setup. [‡]Since SAT consists of only 32 multiple-choice questions, we report AVG score without SAT to account for abnormal cases. The best results are in **bold**, while second-best ones are underlined.

| Model | Method | GSM8K | MATH | SVAMP | ASDiv | MAWPS | TAB* | MQA | MMLU* STEM | SAT* | AVG | AVG† (w/o *) | AVG‡ (w/o SAT) |
|---|---|---|---|---|---|---|---|---|---|---|---|---|---|
| TinyLlama ($S$) | - | 2.7 | 3 | 10.9 | 17.9 | 20.5 | 12.5 | 13.9 | 16.4 | 21.9 | 13.3 | 11.5 | 12.2 |
| TinyLlama-CPT | - | 6.8 | 4.2 | 22 | 36.4 | 47.1 | 16.5 | 12.3 | 23.2 | 15.6 | 20.5 | 21.5 | 21.1 |
| *Teacher w/ Same Vocabulary* | | | | | | | | | | | | | |
| Rho-1 | - | 7.1 | 5 | 23.5 | 41.2 | 53.8 | - | 18 | - | - | - | 24.8 | - |
| $S$ + TinyLlama-CPT | KLD | 6.8 | 5.6 | 22.7 | 37.1 | 49.7 | 17.9 | 12.1 | 23.5 | 15.6 | 21.2 | 22.3 | 21.9 |
| $S$ + TinyLlama-CPT | Ours | 7.4 | 4.6 | 21.7 | 37.7 | 48.0 | 16.7 | 13.0 | 22.5 | 25.0 | 21.8 | 22.1 | 21.5 |
| $S$ + Llemma | KLD | 6.9 | 4.2 | 23.3 | 37.7 | 49.9 | 17.2 | 12.7 | 21.9 | 18.8 | 21.4 | 22.5 | 21.7 |
| $S$ + Llemma | Ours | 8.1 | 5.2 | 21.9 | 38.1 | 50.1 | 21.0 | 13.9 | 24.0 | 34.4 | 24.1 | 22.9 | 22.8 |
| *Teacher w/ Different Vocabulary* | | | | | | | | | | | | | |
| $S$ + Mistral-ProXMath | ULD | 6.0 | 5.4 | 20.9 | 36.4 | 46.7 | 16.7 | 11.2 | 21.1 | 31.2 | 21.7 | 21.1 | 20.6 |
| $S$ + Mistral-ProXMath | Ours | 8.6 | 6.2 | 22.6 | 39.5 | 51.2 | 21.7 | 17.3 | 25.6 | 25.0 | 24.2 | 24.2 | 24.1 |
| $S$ + DeepSeekMath | ULD | 6.3 | 4.8 | 22.4 | 36.8 | 46.0 | 16.6 | 12.2 | 22.4 | 31.2 | 22.1 | 21.4 | 20.9 |
| $S$ + DeepSeekMath | Ours | 9.5 | 6.2 | 23.1 | 41.6 | 53.3 | 22.6 | 15.9 | 25.6 | 18.8 | 24.1 | 24.9 | 24.7 |
| $S$ + Qwen2.5-Math | ULD | 5.8 | 3.6 | 21.3 | 36.1 | 47.1 | 18.0 | 11.7 | 22.4 | 34.4 | 22.3 | 20.9 | 20.8 |
| $S$ + Qwen2.5-Math | Ours | 9.9 | 5.4 | 25.6 | 42.2 | 54.1 | 20.8 | 17.4 | 26.9 | 31.2 | 25.9 | 25.8 | 25.3 |
| $S$ + DeepSeekMath-RL | ULD | 6.7 | 4.6 | 20.8 | 36.1 | 45.8 | 17.9 | 11.2 | 19.5 | 31.2 | 21.5 | 20.9 | 20.3 |
| $S$ + DeepSeekMath-RL | Ours | 10.8 | 7.2 | 27.3 | 45.9 | 59.6 | 22.6 | 19.1 | 28.1 | 21.9 | 26.9 | 28.3 | 27.6 |
| $S$ + Qwen2.5-Math-Inst | ULD | 6.7 | 4.6 | 22.6 | 36.8 | 46.9 | 17.3 | 13.2 | 22.4 | 31.2 | 22.4 | 21.8 | 21.3 |
| $S$ + Qwen2.5-Math-Inst | Ours | 11.3 | 7.6 | 28.9 | 46.8 | 60.7 | 22.5 | 20.5 | 30.3 | 40.6 | **29.9** | **29.3** | **28.6** |

ume of tokens are processed, the impact of vocabulary mis-alignment becomes increasingly pronounced. This highlights the inherent limitations of probabilistic distance-based logit alignment while emphasizing the necessity and effectiveness of a fine-grained, token-level alignment approach.

**5.2. Scalability with Different Teachers.**

As illustrated in Figure 5, VocAgnoLM demonstrates consistent performance improvements when leveraging stronger teacher models, showing a clear correlation with teacher quality. Compared to ULD (Boizard et al., 2025), VocAgnoLM achieves significantly greater performance gains and effectively follows the performance trends of the teacher model. Notably, even though the strongest teacher model, Qwen2.5-Math-Instruct (Yang et al., 2024), has the lowest vocabulary overlap ratio (6.32% in Figure 1) with the student vocabulary, VocAgnoLM still effectively transfers more knowledge and achieves superior performance over logit distribution-based guidance. Additionally, when using DeepSeekMath (Shao et al., 2024) as the teacher model, VocAgnoLM demonstrates competitive performance against Rho-1 (Lin et al., 2024). Further mapping and guidance examples across different teacher models are provided in Appendix C.

## 6. Analysis

### 6.1. Importance of fine-grained sequence alignment

In this section, we extend the preliminary study presented in Section 2 to analyze the significance of Token-level Lexical Alignment's fine-grained, token-level mapping. Following the Section 2, we compare the effectiveness of teacher guidance under two alignment strategies: coarse-grained alignment using chunking and fine-grained alignment by Token-level Lexical Alignment. For this analysis, we employ the Mistral-ProXMath 7B (Zhou et al., 2024) as the teacher and train on 5B tokens from OpenWebMath (Paster et al., 2024), applying the same teacher loss-based guidance detailed in Section 3.2.

In Table 2, we observe performance changes by increasing the number of chunks from 8 to 64. As shown in Figure 6, the performance improves as the chunks become more fine-grained at first. However, when the number of chunks reaches 64, performance begins to degrade, and becomes even worse than the results where chunks are selected randomly instead of using teacher guidance. According to the preliminary study, we measure the character-level IoU and IoS between teacher and student chunks. We find that as the number of chunks increases from 32 to 64, IoU decreases sharply, and this drop corresponds to the significant

Table 2: Performance Comparison of Chunking and Token-level Lexical Alignment.

| Guidance | Num Chunk | GSM8K | MATH | SVAMP | ASDiv | MAWPS | TAB* | MQA | MMLU* STEM | SAT* | AVG | AVG† (w/o *) | AVG‡ (w/o SAT) |
|---|---|---|---|---|---|---|---|---|---|---|---|---|---|
| *Chunking Alignment* | | | | | | | | | | | | | |
| Random | 8 | 3.6 | 3.2 | 19 | 30.3 | 39.5 | 15.7 | 10.6 | 20 | 25 | 18.5 | 17.7 | 17.7 |
| Teacher | 8 | 4.9 | 3.4 | 18.9 | 29.7 | 41.3 | 16.5 | 11.5 | 19.5 | 21.9 | 18.6 | 18.3 | 18.2 |
| Random | 16 | 3.6 | 2.4 | 18.3 | 29.7 | 39.0 | 16.4 | 11.0 | 19.3 | 25.0 | 18.3 | 17.3 | 17.5 |
| Teacher | 16 | 5.1 | 3.8 | 18.8 | 30.7 | 40.8 | 17.7 | 10.5 | 20.0 | 21.9 | 18.8 | 18.3 | 18.4 |
| Random | 32 | 4.2 | 2.6 | 19.5 | 30.2 | 39.6 | 16.6 | 11.5 | 19.1 | 25.0 | 18.7 | 17.9 | 17.9 |
| Teacher | 32 | 5.2 | 4.0 | 18.5 | 30.3 | 41.6 | 17.2 | 11.3 | 20.3 | 21.9 | 18.9 | 18.5 | 18.6 |
| Random | 64 | 3.9 | 3.0 | 19.0 | 29.7 | 38.5 | 16.0 | 11.5 | 19.1 | 25.0 | 18.4 | 17.6 | 17.6 |
| Teacher | 64 | 4.2 | 3.6 | 17.7 | 29.7 | 38.5 | 17.0 | 11.4 | 20.3 | 18.8 | 17.9 | 17.5 | 17.8 |
| *Token-level Lexical Alignment* | | | | | | | | | | | | | |
| Teacher | - | 5.3 | 5.4 | 18.0 | 30.8 | 42.8 | 17.1 | 12.4 | 21.8 | 28.1 | **20.2** | **19.1** | **19.2** |

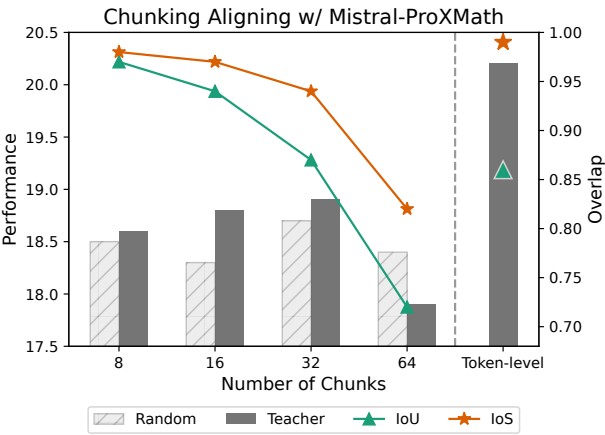

Figure 6: Correlation Between Performance and Sequence Alignment. Chunking alignment initially improves performance as the granularity increases, but performance sharply declines when the overlap (Char-level IoU and IoS) decreases significantly. Token-level Lexical Alignment maintains high IoS and achieves superior performance with fine-grained alignment.

performance degradation. In contrast, Token-level Lexical Alignment demonstrates superior performance with a higher IoU. Due to intrinsic token differences between the teacher and student at fine-grained granularity, we further compare character-level Intersection over Student (IoS). Token-level Lexical Alignment achieves 100% overlap, indicating that all student tokens are covered by teacher tokens, ensuring precise fine-grained teacher guidance.

## 6.2. How to deal with unmapped student tokens?

For student tokens that are not lexically mapped to any teacher tokens (e.g., special tokens), there exists ambigu-

ity in how they should be handled, as they do not receive any guidance from the teacher model. To address this, we compare three strategies: *Mean*, *Exclude*, and *Include*.

The *Mean* strategy average all teacher token losses within a batch to maintain a consistent guidance scale in Equation (4). The *Exclude* strategy discards all unmapped student tokens, assuming they lack semantic information. In contrast, the *Include* strategy trains all unmapped tokens, expecting the role of special tokens.

As shown in Table 3, the *Include* strategy achieve the best performance. Notably, the *Exclude* strategy results in a significant performance degradation, highlighting the critical role of special tokens in continual pretraining. This suggests that unmapped tokens, such as start and end tokens, have already been trained as critical elements during pretraining, providing explicit signals to effectively interpret large amounts of text in the corpus (Devlin et al., 2019; Newman et al., 2020; Yue et al., 2024).

## 6.3. How to aggregate multiple-mapped teacher tokens?

In Equation (2), various aggregation functions ($\Phi$) can be considered to handle the mapping of multiple teacher tokens to a single student token. Intuitively, student tokens mapped to teacher tokens exhibiting high loss values are discarded. To effectively filter out abnormal teacher tokens, we evaluate two aggregation strategies: *Max* and *Mean*. As discussed in Section 6.2, the *Include* strategy is consistently applied for unmapped student tokens in these experiments.

As shown in Table 3, when trained on a dataset of 2B tokens, the performance difference between the two strategies was minimal. However, when trained for a longer duration on 15B tokens, the *Max* strategy outperformed *Mean*, resulting in an approximately 1.3% improvement in AVG (w/o SAT).

Table 3: Performance Comparison by Unmapped and Multi-Mapped Token Strategies.

| Function | Train Tokens | GSM8K | MATH | SVAMP | ASDiv | MAWPS | TAB* | MQA | MMLU* STEM | SAT* | AVG | AVG[†] (w/o *) | AVG[‡] (w/o SAT) |
|---|---|---|---|---|---|---|---|---|---|---|---|---|---|
| *Unmapped Student Tokens Strategy* | | | | | | | | | | | | | |
| Mean | 2B | 3.7 | 4.6 | 16.2 | 24.5 | 32.2 | 14.6 | 10.1 | 15.8 | 25.0 | 16.3 | 15.2 | 15.2 |
| Exclude | 2B | 1.4 | 3.8 | 2.8 | 7.6 | 9.3 | 4.1 | 4.0 | 15.6 | 12.5 | 6.8 | 4.8 | 6.1 |
| Include | 2B | 3.6 | 3.6 | 18.2 | 25.7 | 35.9 | 14.1 | 12.7 | 16.5 | 18.8 | **16.6** | **16.6** | **16.3** |
| *Multi-mapped Teacher Tokens Aggregation* | | | | | | | | | | | | | |
| Include+Mean | 2B | 3.6 | 3.6 | 18.2 | 25.7 | 35.9 | 14.1 | 12.7 | 16.5 | 18.8 | 16.6 | 16.6 | 16.3 |
| Include+Max | 2B | 3.6 | 4.0 | 17 | 26.2 | 35.9 | 14.9 | 12.5 | 16.0 | 18.8 | 16.5 | 16.5 | 16.3 |
| Include+Mean | 15B | 8.3 | 4.6 | 24.0 | 40.2 | 53.9 | 20.8 | 13.0 | 25.5 | 28.1 | **24.3** | 24.0 | 23.8 |
| Include+Max | 15B | 8.6 | 6.2 | 22.6 | 39.5 | 51.2 | 21.7 | 17.3 | 25.6 | 25.0 | 24.2 | **24.2** | **24.1** |

# 7. Related Works

## 7.1. Cross Vocabulary Alignment

Cross-vocabulary alignment has emerged as a critical research area due to tokenization discrepancies between different LLMs. This alignment has been applied to various downstream tasks, such as knowledge distillation (Boizard et al., 2025; Zhang et al., 2024b; Cui et al., 2025), model ensemble (Xu et al., 2024; Huang et al., 2024; Yu et al., 2024), cross-lingual transfer (Dobler & De Melo, 2023). Previous studies primarily rely on matching-based methods (Fu et al., 2023; Wan et al., 2024), optimal transport (Boizard et al., 2025; Cui et al., 2025) , vocabulary mapping matrices (Xu et al., 2024; Huang et al., 2024; Yu et al., 2024) or cross-model attention mechanism (Zhang et al., 2024b). While these approaches have shown effectiveness in specific contexts, they often lack generality when applied to causal language modeling. In contrast, we propose Token-level Lexical Alignment, a simple yet effective method that leverages the guidance of teacher models to enhance general causal language modeling. Furthermore, inspired by Xie et al. (2024); Xu et al. (2025), we conduct a comprehensive study on the effectiveness of alignment at various levels of granularity for language modeling.

## 7.2. Teacher Guided Language Modeling

**Soft label Guidance.** Soft label guidance, a.k.a knowledge distillation, relies on a large teacher model to transfer its logits distribution to a student model. A common approach is to use the logits distribution of teacher model to guide the student model through well-designed optimization objectives (Hinton et al., 2015; Gu et al., 2024a; Agarwal et al., 2024; Boizard et al., 2025), particularly for specific downstream tasks. However, these approaches typically require the the teacher and student models to share the same vocabulary, which limits their applicability when transferring knowledge across models with different tokenization schemes. MiniPLM (Gu et al., 2024b) enables cross-vocabulary distillation through an offline strategy based on difference sampling. In contrast, VocAgnoLM supports both offline and online strategy, offering greater flexibility in the choice of pretraining strategies. Moreover, while MiniPLM performs distillation at the instance-level, VocAgnoLM focuses on token-level, allowing for more fine-grained guidance.

**Hard label Guidance.** Another guidance is training the student model using text generated by teacher model (a.k.a hard labels). (Kim & Rush, 2016; Hsieh et al., 2023; Peng et al., 2023; Zhou et al., 2024; Maini et al., 2024; Gunasekar et al., 2023). However, this method involves substantial overhead, as it requires constructing new training corpora for pretraining. An alternative strategy is leveraging teacher models for data sample selection (Albalak et al., 2024), using heuristic filtering, classifier or perplexity-based filtering, domain specific filtering, deduplication. Irreducible Curriculum (Fan & Jaggi, 2023) extends this idea through batch selection methods for pretraining. However, these coarse-grained approach lack the granularity required for more precise guidance. Rho-1 (Lin et al., 2024) addresses fine-grained token-level data selection. Nevertheless, the requirement of same teacher and student vocabulary limits the capability to transfer knowledge from various teacher models with different vocabulary. In this work, we propose a novel method that enables token-level guidance without requiring the creation of new corpora or identical vocabularies between teacher and student models. By facilitating token-level alignment with any pre-trained or newly released model, our approach significantly enhances the flexibility and applicability of teacher-guided pretraining.

# 8. Conclusion

In this work, we propose Vocabulary-agnostic Teacher Guided Language Modeling (VocAgnoLM), a method for training student models with strong teacher models regardless of vocabulary differences. We identify two key

challenges: *token sequence mismatch* and *logit distribution divergence*, and introduce Token-level Lexical Alignment along with teacher loss-based guidance to address these issues. Our results demonstrate that student performance improves in proportion to the teacher model's capabilities, effectively overcoming vocabulary mismatches. Furthermore, we highlight the significance of token alignment by analyzing the impact of sequence misalignment caused by differences in granularity. Our findings suggest that precise token correspondence plays a crucial role in teacher guidance, providing insights for future research on the effective utilization of teacher models in a vocabulary-agnostic setting.

## Limitations

While we demonstrate VocAgnoLM on TinyLlama 1.1B (Zhang et al., 2024a) using the 15B OpenWeb-Math (Paster et al., 2024), VocAgnoLM is designed to be broadly applicable across different models and datasets. Due to computational constraints, we present a case study demonstrating its effectiveness in continual pretraining on a mathematical domain corpus, leaving further large-scale validation for future work.

## Impact Statement

This paper presents work whose goal is to advance the field of Machine Learning by improving pretraining for student models using a vocabulary-agnostic teacher model. Our study is conducted on OpenWebMath, a publicly available dataset, and focuses solely on mathematical reasoning. We do not foresee significant ethical concerns but acknowledge that language models may still inherit biases from training data, which should be considered in broader applications.

## Acknowledgements

This research was supported by the MSIT (Ministry of Science, ICT), Korea, under the Global Research Support Program in the Digital Field program (RS-2024-00436680) supervised by the IITP (Institute for Information & Communications Technology Planning & Evaluation). And, this project was also supported by Microsoft Research Asia.

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

Table 4: Performance of Teacher Models on Math Evaluation Suite.

| Teacher Model | Num. Vocabs | GSM8K | MATH | SVAMP | ASDiv | MAWPS | TAB | MQA | MMLU STEM | SAT | AVG |
|---|---|---|---|---|---|---|---|---|---|---|---|
| Llemma | 32K | 38.8 | 17.2 | 56.1 | 69.1 | 82.4 | 48.7 | 41.0 | 45.4 | 59.4 | 50.9 |
| Mistral-ProXMath | 32K | 51.0 | 22.4 | 64.9 | 72.9 | 89.2 | 49.8 | 53.0 | 54.2 | 75.0 | 59.2 |
| DeepSeekMath | 100K | 64.1 | 34.2 | 74.0 | 83.9 | 92.4 | 63.4 | 62.4 | 56.4 | 84.4 | 68.4 |
| DeepSeekMath-RL | 100K | 86.2 | 50.2 | 87.7 | 91.1 | 96.6 | 64.9 | 56.9 | 24.5 | 15.6 | 63.7 |
| Qwen2.5-Math | 150K | 85.8 | 57.4 | 88.2 | 91.7 | 96.3 | 67.3 | 75.9 | 69.4 | 93.8 | 80.6 |
| Qwen2.5-Math-Instruct | 150K | 88.3 | 74.0 | 90.3 | 90.8 | 93.8 | 81.6 | 81.0 | 65.2 | 90.6 | 84.0 |

## A. Teacher Model Details

**Vocabulary Details.** The tokenization schemes of the teacher models vary significantly. Mistral-ProXMath (Zhou et al., 2024) adopts Byte Pair Encoding (BPE) with a vocabulary size of 32,000 tokens. DeepSeekMath (Shao et al., 2024) employs Byte-level Byte Pair Encoding (BBPE) with a larger vocabulary size of 100,000 tokens, while Qwen2.5-Math (Yang et al., 2024) utilizes BPE with the largest vocabulary size of 150,000 tokens among the teacher models. These variations highlight the diversity of vocabularies for each teacher models.

**Teacher Performances.** In Table 4, we report the performance of teacher models to compare the impact of teacher guidance based on their capabilities.

## B. Further Implementation Details

**KLD.** Following the setup outlined by Song et al. (2020) and Gu et al. (2024a), we combine the KL-Divergence loss calculated between the teacher and student output distribution $(p_i^S, p_i^T)$ with the cross-entropy loss of the student model, using the same weighting ratio.

$$\mathcal{L}_{\text{KLD}}(x^S, x^T) = -\sum_{i=1}^{|x^S|} \log P\left(x_i^S \mid x_{<i}^S\right) + \sum_{i=1}^{|x^S|} \text{KL}\left(p_i^S \| p_i^T\right) \tag{5}$$

**ULD.** Boizard et al. (2025) proposes a Universal Logit Distance (ULD) loss, as shown in Equation (6), to measure the logit distribution distance between teacher and student with different vocabularies. ULD utilizes Wasserstein distance (WD) to calculate the distance of teacher and student output distribution $(p_i^S, p_j^T)$. While Boizard et al. (2025) primarily focus on downstream tasks, we explore the hyperparameter $\lambda$ values [0.3, 0.5, 1.0, 1.5] to determine those most suitable for continual pretraining on OpenWebMath (Paster et al., 2024). Based on the results in Figure 7b, we adopt $\lambda = 0.5$ in this study. For the further details, please refer to Boizard et al. (2025).

$$\mathcal{L}_{\text{ULD}}(x^S, x^T) = -\sum_{i=1}^{|x^S|} \log P\left(x_i^S \mid x_{<i}^S\right) + \lambda \cdot \sum_{i=1}^{|x^S|} \sum_{j=1}^{|x^T|} \text{WD}\left(p_i^S \| p_j^T\right). \tag{6}$$

**Llemma.** Llemma (Azerbayev et al., 2024) adopts the Llama-2 (Touvron et al., 2023) vocabulary, similar to TinyLlama (Zhang et al., 2024a), with the addition of 16 noise tokens. However, we observe that the cumulative probability of these tokens is overall negligible, below $1 \times 10^{-7}$. Consequently, this study disregards the probabilities of noise token and considers Llemma (Azerbayev et al., 2024) as a teacher model that shares the same vocabulary as TinyLlama (Zhang et al., 2024a).

**Threshold.** We explore the impact of the top-k threshold on a 2B-token subset of the corpus, evaluating thresholds ranging from 40% to 80%, following the settings in Lin et al. (2024). Based on the results in Figure 7a, we select 40% threshold for our experiments.

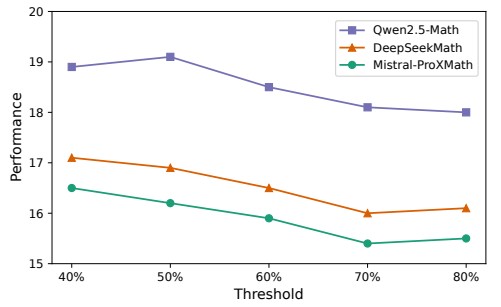

(a) Performance of student model guided by various teacher models based on different Top-K thresholds

(b) Performance analysis of $\lambda$ on ULD (Boizard et al., 2025).

Figure 7: Performance Comparison for Hyperparameter Searching.

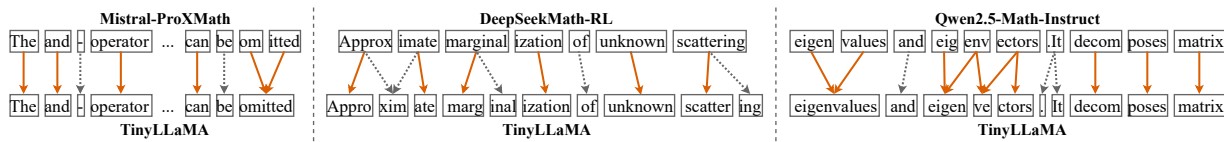

Figure 8: Examples of teacher guidance demonstrating how teacher models influence the training of TinyLLaMA (Zhang et al., 2024a). Orange solid arrows represent the guidance provided by the teacher model, while gray dashed arrows indicate tokens that were not included in the top-k threshold by teacher guidance.

Table 5: Performance Comparison Using Teacher Models with Comparable Performance: Same vs. Different Vocabularies.
[†][‡]Average scores for abnormal cases, following the setup in Table 1.

| Model | Vocab w/ Student | Method | AVG | AVG[†] (w/o *) | AVG[‡] (w/o SAT) |
|---|---|---|---|---|---|
| *Teacher Performance* | | | | | |
| MetaMath-Llemma | Same | - | | 52.2 | 62.1 | 57.2 |
| Mistral-ProXMath | Different | - | | 59.2 | 58.9 | 57.2 |
| *Student (S) Distilled Performance* | | | | | |
| $S$ + MetaMath-Llemma | Same | KLD | 14.2 | 14.3 | 14.4 |
| $S$ + Mistral-ProXMath | Different | Ours | 16.6 | 16.6 | 16.3 |

## C. Teacher Guidance Examples

Figure 8 illustrates how the teacher model effectively guides important tokens. Even when multiple teacher tokens are mapped to a single student token (1:N mapping), the teacher provides meaningful alignment. Furthermore, less important tokens (such as "-" and ".It" in Figure 8), do not pass the top-k threshold, highlighting the effectiveness of the guidance process.

## D. Additional Comparison with Other Baselines

**Comparable Teacher Models with Same vs. Different Vocabularies.**    To assess the effectiveness of our approach, we conduct an additional experiment on 2B tokens by comparing it against a standard distillation setup using two teacher models with comparable performance: MetaMath-Llemma-7B (Yu et al., 2023), which shares the same vocabulary as the student model, and Mistral-ProXMath-7B (Zhou et al., 2024), which uses a different vocabulary. As shown in Table 5, VocAgnoLM performs better even when using a teacher model with a different vocabulary, despite both teacher models achieving the same performance on the AVG[‡] metric.

Table 6: Comparison of Scaled Continual Pretraining Baseline and VocAgnoLM.

| Model | AVG (w/o SAT) |
|---|---|
| CPT | 13.6 |
| CPT (lr /= 0.4) | 15.3 |
| $S$ + Mistral-ProXMath | 16.3 |
| $S$ + DeepSeekMath | 17.3 |
| $S$ + Qwen2.5-Math | 18.8 |

**Scaled Continual Pretraining Baseline.** To evaluate the utility of teacher loss based guidance, we compare our method against a continual pretraining baseline (CPT) on 2B tokens, where the training signal is uniformly amplified. Specifically, as we apply a top-40% threshold based on teacher guidance (see Appendix B), we rescale the learning rate by a factor of $\frac{lr}{0.4} = 2.5\times$ to match the overall signal strength delivered to the important tokens. As shown in Table 6, our method, using various teacher models, outperforms CPT (lr/ = 0.4). Although CPT delivers a similar amount of signal strength by increasing the learning rate, it also amplifies the effect of unimportant tokens. These results highlight the effectiveness of our teacher guidance, and provide a way to estimate the impact of various teacher models in terms of CPT-equivalent training scale.

