# OpenReview forum: "Overcoming Vocabulary Mismatch: Vocabulary-agnostic Teacher Guided Language Modeling"
_ICML.cc/2025/Conference — ICML 2025 poster_

### Official Review · Reviewer_ZCju · 2025-03-12

**Overall Recommendation:** 2

**Summary:**

This paper proposes a Vocabulary-agnostic Teacher Guided Language Modeling for guiding the training of smaller student models by large teacher models. This method tries to bridge the gap caused by vocabulary mismatch in different models. The proposed approach comprises two key components: Token-level Lexical Alignment and Teacher Guided Loss, both of which contribute to the enhancement of performance.

**Claims And Evidence:**

The authors argue that the perceived vocabulary mismatch is, in fact, a mismatch of tokens extracted by different models. Presenting this as a vocabulary mismatch appears to introduce a misleading notion.

**Essential References Not Discussed:**

No

**Experimental Designs Or Analyses:**

Yes, I have checked the validity of experimental designs, and they are  acceptable.

**Methods And Evaluation Criteria:**

The proposed methods and evaluation criteria are reasonable.

**Other Comments Or Suggestions:**

No

**Other Strengths And Weaknesses:**

**Strengths**

The presentation of the paper is good, with each section being clear. The focused problem is well-articulated, with an in-depth analysis and preliminary tiny experiments to illustrate it.

The suggested method achieves effective performance improvements.

The scalability with different teachers has also been sufficiently validated.

**Weaknesses**

The contributions of this paper are insufficient. Firstly, the Token-level Lexical Alignment appears to be an engineering operation, and the theoretical analysis of the effectiveness of the proposed mapping is lacking. Secondly, the technical contribution of weighting different tokens is not substantial enough.

**Questions For Authors:**

Please refer to the Weaknesses.

**Relation To Broader Scientific Literature:**

There's nothing to add

**Theoretical Claims:**

The authors lack corresponding theoretical analysis, and the proposed solution tends to be more engineering-oriented.

---

> ### Author Rebuttal · Authors · 2025-04-01
>
> Thank you for the reviewer’s helpful feedback. We are pleased that the reviewer recognized the motivation and analysis of our work. We provide our response as below.
>
> ### **Clarification of the “Vocabulary Mismatch” Notion.**   (Claims And Evidence)
> We appreciate the chance to clarify our terminology and the motivation behind it.
>
> - **Tokenizer Differences Originate from "Vocabulary" Choices:**
> The issue of "tokens extracted by different models" fundamentally stems from the fact that these models use different tokenizers. When building a tokenizer, the primary goal is to construct a vocabulary. Since different model series typically start with tokenizers based on distinct vocabularies, we believe that the fundamental reason for the observed mismatch lies in the vocabulary itself. Thus, we refer to this problem as a *vocabulary mismatch*.
>
> - **"Vocabulary" Mismatch Leads to "Token" Sequence and Logit Mismatches:**
> The consequences of vocabulary mismatch manifest as differences in token sequences and divergences in logit distributions. Since the token sequence is determined by the tokenizer’s vocabulary, we treat “token sequence mismatch” as a downstream result of vocabulary mismatch. Moreover, in transformer-based models, the vocabulary size directly determines the dimensionality of the final logit layer. Therefore, “logit distribution divergence” is more precisely attributed to differences in vocabulary, rather than just token-level differences.
>
> - **Terminological Consistency with Prior Work:**
> A large body of prior work [1,2,3] has used terms in a way that implicitly refers to mismatches in vocabulary. In this context, using the term *vocabulary mismatch* aligns with established usage in the field. The suggestion that it is misleading may in fact run counter to prevailing terminology practices in related literature.
>
> [1] *Cui et al., Multi-level optimal transport for universal cross-tokenizer knowledge distillation on language models. In AAAI 2025.*
>
> [2] *Boizard et al., Towards cross-tokenizer distillation: the universal logit distillation loss for LLMs. TMLR, 2025.*
>
> [3] *Xu et al., Bridging the gap between different vocabularies for llm ensemble. In NAACL 2024.*
>
>
> ### **Highlighting Our Contribution on Vocabulary Mismatch in Pretraining.**   (Weaknesses)
> We would like to highlight that **addressing *vocabulary mismatch* in the pretraining stage** is one of the main contributions of our work.
> While knowledge distillation during pretraining is a well-established technique, in practice, vocabulary mismatch often limits the use of diverse teacher models.
> To our best knowledge, this aspect remains underexplored in the pretraining literature, and we believe **this is the first work** to directly address vocabulary mismatch in this context.
> VocagnoLM offers **a simple yet effective** solution to this problem, with practical benefits as follows:
> - **Practicality of Token-Level Lexical Alignment**
>   - While our token-level lexical alignment may not constitute a novel theoretical contribution, we argue that this is precisely what makes it a **practically effective** solution to the vocabulary mismatch problem. Our proposed method performs lexical alignment based on character offsets, which theoretically ensures that every student token must be contained within one or more teacher tokens. In practice, as shown in Figure 6, we observe 100% Intersection-of-String (IoS), demonstrating that two token sequences can be efficiently and accurately aligned using only a simple approach.
>   - Furthermore, this practical nature allows for both *online mapping* during training and *offline mapping* during preprocessing. This flexibility is particularly advantageous in large-scale pretraining scenarios, where optimization at the training stage is crucial.
>   - We highlight the simplicity and efficiency of our mapping function, which supports various weighting options in the aggregation step. While we only explore a simple aggregation function in this paper, we leave the exploration of more advanced variants as future work.
>
> - **Advantages of Loss-Based Teacher Guidance for Logit Distribution Divergence**
>   - Even after resolving the token sequence mismatch, the issue of logit distribution divergence remains. Traditional KL divergence cannot adequately address this, and existing distance-based methods such as ULD, can incur additional information loss during alignment.
>   - In contrast, our approach is based on the teacher’s loss, which allows us to circumvent the problem arising from the dimensional mismatch between vocabularies.
>
> In summary, we **decompose the issues caused by vocabulary mismatch into two sub-problems (token sequence mismatch / logit distribution divergence)** and design separate solutions for each. We demonstrate that applying these techniques in the pretraining stage leads to effective improvement.

---

> > ### Comment · Reviewer_ZCju · 2025-04-09
> >
> > Thank you for the author’s response. They partally address my concerns. I also reviewed the comments from other reviewers, and I believe that the contribution of this manuscript is not sufficient for publication in ICML. I maintain my original rating.

---

### Official Review · Reviewer_bj8e · 2025-03-13

**Overall Recommendation:** 3

**Summary:**

The paper addresses the challenge of vocabulary mismatches between teacher and student language models during knowledge distillation. I believe this is a well-motivated and important topic since it is difficult to do the logits-level distillation between student and teacher models with different tokenizers. To overcome this, the authors propose the method that consists of two main components:
• Token-level Lexical Alignment: A procedure to map student tokens to the corresponding teacher tokens using character-level offsets, thereby aligning token sequences despite different vocabularies.
• Teacher Guided Loss: A reweighting scheme that leverages the teacher’s token-level loss as guidance for training the student, overcoming divergences in logit distributions.
The approach is validated on a math-focused pretraining corpus (OpenWebMath) and evaluated on a suite of mathematical reasoning benchmarks, showing significant performance improvements—especially when using teacher models with very low vocabulary overlap.

**Claims And Evidence:**

I believe the claims (the motivation and effectiveness of the design) are generally well-supported by the experiments. The potential questions are listed in the other section. The experimental results indicate that VocAgnoLM improves performance by up to 46% compared to naive continual pretraining and outperforms logit distribution alignment baselines (e.g., KLD and ULD). Detailed ablation studies are provided to show how choices in token alignment granularity, handling of unmapped tokens, and aggregation strategies affect performance.

**Essential References Not Discussed:**

There are more existing literatures considering the knowledge distillation (not necessarily the cross-tokenization KD), such as the KD for pretraining exploration (https://arxiv.org/pdf/2410.16215) and the Nvidia minitron (https://arxiv.org/pdf/2407.14679). It is good to include more previous papers on KD to further emphasize the importance of the KD-based pretraining.

**Experimental Designs Or Analyses:**

It is appreciated that the author compares the performance of VocAgnoLM to KLD-based and ULD-based distillation, and they also provide ablation study to demonstrate the effectiveness of the design.

However, the evaluation is not comprehensive enough to demonstrate the bounds of the effectiveness of the proposed methods. We does want to understand the performance gain/loss of the proposed methods but the comparison seems not to be very fair. It would be great if the following two comparisons are provided: 1) learn from the logits of advanced models (e.g. qwen-math) vs. learn from their generated tokens. 2) Two similar student models (with different tokenizers) learn from the same teacher (or teachers with similar performance) - one with identical tokenizer vs one with different tokenizer. By this two experiments, we can understand the bound/limit of the proposed methods, which can be very helpful.

Also, the computational complexity can be high, it would be great if more comprehensive and explicit data points can be provided.

**Methods And Evaluation Criteria:**

The proposed methods make sense to me while they are somehow heuristic and lack theoretical support for their effectiveness.

a) Token-level Lexical Alignment: 1) Utilizes character-level offsets from both teacher and student tokenizers. 2) Employs binary search techniques to establish a one-to-many mapping for each student token.
b) Teacher Guided Loss: 1) Computes the loss for a student token and aggregates the losses of its corresponding teacher tokens. 2) Applies a top-k threshold strategy to reweight the importance of each student token based on the discrepancy between the student and teacher losses.

Although the evaluation is mainly concentrated on the math/reasoning tasks, I believe they are representative. They compare performance on multiple math reasoning benchmarks (e.g., GSM8K, MATH, SVAMP, ASDiv, MAWPS, and others). Comparisons against standard distillation approaches (KLD and ULD) are also provided. Ablation studies that assess the impact of different alignment granularities and token handling strategies, which demonstrate the effectiveness of the design. The weakness of the evaluation is mentioned in the previous section.

**Other Comments Or Suggestions:**

Please kindly refer to the weakness discussed above.

**Other Strengths And Weaknesses:**

* Strength:
Comprehensive Analysis: Extensive ablation studies and comparisons with existing methods (KLD, ULD) strengthen the evidence for the proposed approach.

Practical Impact: By enabling the use of high-performing teacher models regardless of vocabulary, the method broadens the applicability of teacher-guided pretraining, particularly in domain-specific settings like mathematics.

* Weakness:

The methods are heuristic, and we are not sure how much the proposed methods will compromise/influence the KD performance, compared to the standard KD with the same tokenizer.

The topic is of great importance and I will raise my score if concerns can be properly addressed.

**Questions For Authors:**

Please kindly refer to the question in the previous sections.

**Relation To Broader Scientific Literature:**

I believe the proposed methods are important and relevant to general LLM pretraining, especially for knowledge distillation domain. Previous works mainly consider the distillation between models of the same tokenizer. Recent works 'Towards cross-tokenizer distillation: the universal logit distillation loss for LLMs' proposes methods for cross-tokenizer distillation. In this paper, the proposed methods are shown to outperform the existing cross-token KD methods and improve the performance based on KD process.

**Theoretical Claims:**

N/A

---

> ### Author Rebuttal · Authors · 2025-04-01
>
> We sincerely appreciate the reviewer’s insightful comments and valuable suggestions. We are glad that the reviewer acknowledged the importance of the problem and appreciated our comprehensive analysis. Below, we provide our detailed responses to the points raised.
>
> ### **Distinction between "Teacher-generated Knowledge" and "Teacher-Guided Language Modeling"** (Exp Designs or Analyses #1)
> We would first like to clarify that the scope of this work specifically targets the pretraining stage. Unlike (sequence-level) knowledge distillation [1] approaches trained on generated tokens in the fine-tuning stage, it is difficult to define the “generated tokens” of the teacher model during the pretraining stage. In the extreme case, a teacher model could reproduce the same input corpus or generate an entirely new 15B corpus using the teacher model’s internal knowledge.
>
> Besides, training the student model on new knowledge generated by the teacher model differs from our objective of teacher-guided language modeling within a given corpus. However, while such an approach goes beyond the scope of this work, we appreciate the suggestion and believe it as a valuable direction worth exploring further.
>
> [1] *Kim et al., Sequence-Level Knowledge Distillation, In EMNLP 2016.*
>
> ### **Comparison with the standard KD using same tokenizer teacher** (Exp Designs or Analyses #2, Weakness)
>
> We appreciate the reviewer’s suggestion for further experiments. We present an additional experimental result on 2B tokens that offers relevant insight. Specifically, we compare our method against the standard KD approach using two teacher models of comparable performance: MetaMath-Llemma-7B (which shares the same tokenizer as the student model) and Mistral-ProXMath-7B (which uses a different tokenizer).
>
> As shown in the table below, our method performs better even when using a teacher model with a different vocabulary, despite both two teacher models achieving the same performance on the AVG(w/o SAT) metric.
>
> | Model                      | Tokenizer w/ Student | Method | AVG  | AVG (w/o *) | AVG (w/o SAT) |
> |---------------------------|----------------------|--------|------|-------------|----------------|
> | ***Teacher Model Performance*** |||||
> | MetaMath-Llemma-7B        | Same                 | -      | 52.2 | 62.1        | 57.2           |
> | Mistral-ProXMath-7B       | Different            | -      | 59.2 | 58.9        | 57.2           |
> | ***Student (S) Model Performance*** |||||
> | S + MetaMath-Llemma-7B    | Same                 | KLD    | 14.2 | 14.3        | 14.4           |
> | S + Mistral-ProXMath-7B   | Different            | Ours   | 16.6 | 16.6        | 16.3           |
>
> We also clarify that our Token-level Lexical Alignment is a deterministic and straightforward solution based on character offset. While simple, it ensures 100% overlap (Figure 6), and plays a critical role in enabling reliable teacher guidance across different tokenizers. We believe that our alignment mechanism contributes to the effectiveness of our method even when compared to standard KD with the same tokenizer, as demonstrated by the results in the table.
>
> ### **Clarification on Computational Overhead.** (Exp Designs or Analyses #3)
> To report the additional computational overhead, we decompose into two components compared to standard KD, Token-level Lexical Alignment and Loss-based Guidance.
>
> - First, the loss-based guidance shares most of its computational operations with standard KLD. When measuring 1M tokens, KLD and loss-based guidance require approximately 15.36 TFLOPs, **showing no significant difference in computation cost.**
> - On the other hand, Token-level Lexical Alignment is a CPU-bound operation, so we report latency instead of FLOPs.. When measuring on a 2048 token sequence using the Mistral-ProXMath teacher model, the alignment step takes approximately 0.047 seconds, averaged on 1000 repeated runs. While this mapping process introduces a small amount of overhead, it can be amortized during preprocessing time. **Especially in pretraining stage, corpus is typically packed to the maximum sequence length, so the mapping process can be efficiently performed during preprocessing time.**
>
> We hope this addresses the reviewer’s concern regarding computational overhead. We’ll include this point in the final version.
>
> ### **Response for "Essential References Not Discussed"**
> Thank you for the new suggestion. Minitron (Muralidharan et al., 2024) has already been cited in the introduction (Line 48-49). We also agree that Pretraining Exploration is a valuable reference that highlights the importance of knowledge distillation during pretraining. In the final version, we will revise the first paragraph of the introduction to emphasize on the importance of KD-based pretraining.

---

> > ### Comment · Reviewer_bj8e · 2025-04-04
> >
> > I appreciate the authors' response and additional experiments. Most of concerns have been addressed and I recommend accepting the paper

---

> > > ### Author Response · Authors · 2025-04-05
> > >
> > > We sincerely appreciate the reviewer’s thoughtful response and are grateful for the recommendation to accept the paper. We're glad that our efforts to address the concerns were well received.

---

### Official Review · Reviewer_HDRn · 2025-03-14

**Overall Recommendation:** 3

**Summary:**

This paper proposes VocAgnoLM, a method to overcome vocabulary mismatch in knowledge distillation for language models. It introduces Token-level Lexical Alignment for precise token mapping and Teacher Guided Loss to adjust training signals. Experiments show up to 46% improvement over baseline methods, enabling effective distillation from stronger teacher models despite vocabulary differences.

**Claims And Evidence:**

In Equation (4), certain tokens are masked during pretraining with the proposed VocAgnoLM method, potentially reducing the effective token count. Given that current scaling laws [1,2] primarily examine the relationship between performance and training computation, this masking mechanism may inherently introduce a scaling disadvantage. To better understand this limitation, more empirical analysis is encouraged, including:

1. Masking Ratios: What are the token masking ratios across different experimental settings in the paper?
2. Scaling Trends: How does the scaling behavior of VocAgnoLM compare to baseline methods in terms of training computation?
3. Generalization to Broader Domains: How does the method perform on general-domain tasks, where more critical tokens may exist beyond the math-focused setting?

[1] Scaling Laws for Neural Language Models.
[2] Training Compute-Optimal Large Language Models.

**Essential References Not Discussed:**

N/A

**Experimental Designs Or Analyses:**

From Equation (4), the proposed method improves upon the baselines by masking out "unimportant tokens," thereby amplifying the supervision signal from the "important tokens." However, a much simpler way to achieve a similar effect is to increase the learning rate uniformly across all tokens, without masking. For instance, if the masking ratio is 20%, raising the learning rate by a factor of 1/(1-0.2) = 1.25x in the standard CPT baseline would result in a comparable signal strength for important tokens as in the knowledge distillation setting. To further validate the effectiveness of Teacher Guided Language Modeling, it would be beneficial to compare VocAgnoLM against this learning rate adjustment baseline.

**Methods And Evaluation Criteria:**

N/A

**Other Comments Or Suggestions:**

N/A

**Other Strengths And Weaknesses:**

N/A

**Questions For Authors:**

In lines 435-439, the authors claim that MiniPLM[1] requires the teacher and student models to share the same vocabulary. However, [1] shows that MiniPLM enables cross-family KD where the vocabulary of the student and teacher models can be different. Is this a mistake in the literature review?

**Relation To Broader Scientific Literature:**

This paper proposes a method to tackle the tokenization mis-matching problem during knowledge distillation.

**Theoretical Claims:**

There is no theoretical claim proposed in the paper.

---

> ### Author Rebuttal · Authors · 2025-04-01
>
> We appreciate the reviewer's thoughtful feedback and constructive suggestion. We address the points raised below.
>
> ### **Impact of top-k threshold (Selected Ratio) on scaling trends.** (Claims and Evidence #1, #2)
> - As discussed in Appendix B and Figure 7a, we explore the effect of the top-k threshold using a 2B-token subset of the corpus, following the experimental setup of Lin et al [2]. Based on this analysis, we adopt a top-k threshold of 40% ( 60% Masking Ratio ) in our study (L253).
> - Consistent with prior work [1,2], we observe that selecting more tokens from a fixed corpus does not necessarily lead to better performance. Importantly, as shown in Figure 7a, the optimal threshold may vary depending on the choice of teacher model. Taking into account both our empirical results in Appendix B and findings from previous study [1,2], we chose the 40% threshold.
>
> ### **Insights from established findings, and highlight our representative tasks.** (Claims and Evidence #3)
> - As mentioned in our limitations section, we agree that VocAgnoLM is designed to be broadly applicable across different models and datasets.. Due to our limited pretraining resources, we focused our efforts on the moderately scaled 15B OpenWebMath corpus. **As reviewer bj8e acknowledged, although our work focuses on math/reasoning tasks, we made every effort to evaluate the model through a diverse set of representative tasks to ensure broad applicability within the domain.**
> - As a complementary perspective, we would like to share an intuition that motivated our design choice. **Several prior works [1,2,3] have provided promising evidence that selective token/dataset training may generalize beyond narrow domains, not only on math domain corpus, but also on general-domain corpora such as ThePile, SlimPajama, StarCoderData.** Motivated by these findings, our study aims to validate whether the proposed method can effectively follow the teacher model’s expertise within a domain, using a moderately scaled math-domain web corpus.
> - We hope that these prior findings, along with our own results (e.g., Figure 5, which shows performance improvement using an instruction-tuned teacher model) can help estimate VocagnoLM’s potential for general-domain performance. We consider extending our experiments to broader general-domain corpora a valuable direction for future work.
>
> [1] *Mindermann et al., Prioritized Training on Points that are learnable, Worth Learning, and Not Yet Learnt, In ICML 2022.*
>
> [2] *Lin et al., RHO-1: Not All Tokens Are What You Need, In NeurIPS 2024.*
>
> [3] *Xie et al., Data Selection for Language Models via Importance Resampling, In NeurIPS 2023.*
>
>
> ### **Comparison with scaled CPT baseline.** (Experimental Designs and Analyses)
> Thank you for the insightful suggestion. Following your suggestion, we conducted an additional experiment on 2B tokens.
>
> In VocAgnoLM, we apply a top 40% threshold based on teacher guidance. To match the signal strength of important tokens, we rescaled learning rate by a factor of lr / 0.4 = 2.5x.
>
> As shown in the table below, our method, using various teacher models, outperforms CPT (lr /= 0.4). Although CPT delivers a similar amount of signal strength by increasing the learning rate, it also amplifies the effect of unimportant tokens. These results support the effectiveness of our teacher guidance, and allow us to estimate the impact of various teacher models in terms of CPT-equivalent training scale.
>
> | Setting                 |  AVG (w/o SAT) |
> |------------------------|----------------|
> | CPT                    | 13.6       |
> | CPT  (lr /= 0.4)    |  15.3           |
> | S + Mistral-ProXMath   | 16.3           |
> | S + DeepSeekMath       | 17.3           |
> | S + Qwen2.5-Math       | 18.8           |
>
>
> ### **Difference with MiniPLM.** (Questions for Authors)
> Thank you for pointing this out. We noticed that the order of the last two sentences in Section 7.2 may have been mistakenly switched. MiniPLM enables cross-vocabulary distillation through an offline KD strategy. However, our method supports **both online and offline KD**, offering more flexibility in the choice of pretraining strategies. Additionally, while MiniPLM performs distillation at the instance-level, our approach focuses on **token-level**, which marks a key difference. We will revise the sentence accordingly and describe this difference in the final version.

---

### Decision · Program_Chairs · 2025-05-01

**Decision:**

Accept (poster)

**Comment:**

This paper tackles a pervasive obstacle in teacher‑student distillation for LLMs: when teacher and student use different tokenizers, their token sequences and logits don’t align. This work offers two pragmatic fixes during pretraining: (1) Token‑level lexical alignment, which uses simple character‑offset matching (binary search over tokenizer spans) to deterministically map each student token to one or more teacher tokens. (2) Teacher‑Guided Loss, which aggregates the teacher’s per‑token losses (over the aligned spans), then applies a top‑k reweighting so the strongest signals drive student updates—avoiding spurious supervision on “unmatched” or low‑importance tokens. Overall, given the practical significance, strong empirical evidence, and the fact that vocabulary mismatch is a known but under‑addressed barrier in pretraining‑stage knowledge distillation, I recommend acceptance of the paper. The authors should carefully address the reviewers' remaining suggestions though, for example, those related to adding a broader set of evaluations.